# A Nonlinear Magnetic Stabilization Control Design for an Externally Manipulated DC Motor: An Academic Low-Cost Experimental Platform

Leonardo Acho 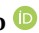

Department of Mathematics, Universitat Politècnica de Catalunya (UPC), 08222 Terrassa, Spain; leonardo.acho@upc.edu

**Abstract:** The main objective of this paper is to present a position control design to a DC-motor, where the set-point is externally supplied. The controller is conceived by using vibrational control theory and implemented by just processing the time derivative of a Hall-effect sensor signal. Vibrational control is robust against model uncertainties. Hence, for control design, a simple mathematical model of a DC-Motor is invoked. Then, this controller is realized by utilizing analog electronics via operational amplifiers. In the experimental set-up, one extreme of a flexible beam attached to the motor shaft, and with a permanent magnet fixed on the other end, is constructed. Therefore, the control action consists of externally manipulating the flexible beam rotational position by driving a moveable Hall-effect sensor that is located facing the magnet. The experimental platform results in a low-priced device and is useful for teaching control and electronic topics. Experimental results are evidenced to support the main paper contribution.

**Keywords:** control; electronics; DC-motors; magnetic manipulation





## 1. Introduction

Direct-Current (DC) motors have been utilized in many control engineering applications [1–3]. These devices can provide a high starting torque and can be controlled over a wide range of (angular) speeds. Hence, DC-Motors are significative in modern automatic industrial processes. A DC-Motor is manufactured such that by varying its terminals voltage, its speed can be regulated [4,5]. On the other hand, position control of DC-Motors is mandatory in some industrial tasks [3,5,6]. From the theoretical point of view, for both position and speed control to DC-Motors, there have been some recent research reports [7–10]. For instance, a PID control with friction compensation by programming a digital controller is presented in [3]. A replacing artificial neural network controller with a PID scheme is developed in [5], where the training data for this intelligent approach is carryout in advanced mathematical software. Moreover, and in the state-of-the-art on designing experimental platforms via a pulse-width-modulation control format, is detailed in [11].

However, in this paper, a recent perspective on position control of a DC-Motor is developed, where the reference command is externally marked by using a moveable Hall-effect magnetic sensor. The control approach is based on commuting the motor speed around a given constant value. Therefore, the controller just requires the speed information of the motor shaft. This information is calculated from a Hall-effect sensor signal. The control design and stability issues are carried out by conceiving a simple mathematical model of the DC-Motor. After that, the obtained control scheme is realized by using operational amplifiers. These electronic components are inexpensive and easy to manipulate in comparison to micro-controllers, which are also low-priced, but needing a programmer, a compiler, and a computer [12]. On the other hand, and due to the switching nature of the controller action, it induces a vibrational behavior on the motor dynamic.

To activate the Hall-effect sensor, a magnet is employed. This magnet is attached to one end of a flexible beam, whereas the other end is fixed to the motor shaft. See Figure 1. Therefore, the control objective consists of manipulating the flexible beam on the motor to track the set-point marked by the moveable Hall-effect sensor, where this sensor is facing the magnet. In literature, there are many controllers including the classical PID method [3,13]. Even so, the main purpose of our controller is to invoke vibrational control theory and to employ just speed information of the motor shaft to control its position in a contactless trend. More precisely, the paper contributions can be stated as follow:

- A contactless vibrational position control design to a DC-Motor.
- A controller realization by just using speed estimation of the DC-Motor.
- A controller based on analog electronics.
- An experimental platform with a moveable magnetic sensor.

From the experimental point of view, a kind of vibrational control design has been applied to DC-Motors, including the case of a hard non-linearity backlash on the control actuator [14]. Therefore, vibrational control is suitable to DC-Motors even when a chattering behavior is presented in the control signal. Obviously, and because of the low-pass filter conduct of the motor, this effect is attenuated. Hence, chattering is not ideally realized on the mechanical part of the DC-Motor.

The use of a Hall-effect sensor in a feedback system for position control of DC-Motors is a cheap technological option [15]. In complement, a low-priced undergraduate control device by using a microcontroller unit for control experimentation is shown in [16]. The cost of this platform is stated as 80 dollars. On the other hand, in [3], a PID control design has been developed by employing the well-known microcontroller Arduino-Uno board, an optical encoder for angular position measurement of the DC-Motor, and an L298 motor driver. And an experimental platform for teaching automatic control can be seen in [17]. Moreover, using high-level software as a human interface to an experimental platform is also possible [17]. Even so, our low-cost experimental platform is a recent approach.

Vibrational control can be viewed as a switching system [18–20]. These systems may respond in finite-time [19]. Therefore, for control design, these switching models are robustly simplifying the control realization [21,22]. This justifies the use of a simple model of the plant just for the control design stage. Therefore, if the plant model is simple, then the control scheme will be too. This due to the control method is based on the plant mathematical modeling. We use this fact to conceive a simple controller as was previously stated. Experimental results support our contribution.

The rest of the paper is outlined as follows. Section 2 shows the control design and its electronic realization by using operational amplifiers. Furthermore, some numerical results are evidenced to appreciate the controller robustness. Section 3 shows the experimental results along with a link to an edited video to support our control approach. Finally, in Section 4 comments are stated.

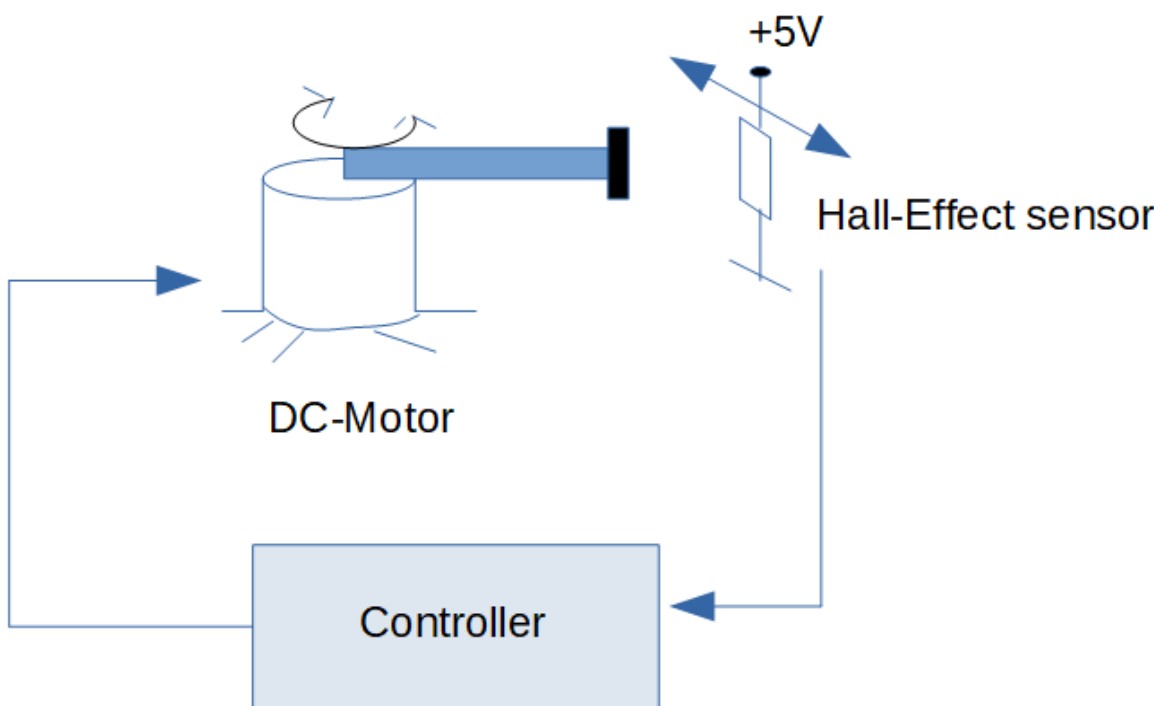

**Figure 1.** A schematic representation of a DC-Motor under control. The magnet, represented in the black drawing, follows the moveable Hall-effect sensor.

## 2. Control Algorithm Design

For control design, let us consider a simple mathematical model of a DC-Motor, an ideal one. Taking into account that the motor speed is proportional to the applied voltage at its terminal connections, it can be captured by:

$$\dot{v}(t) = -u(t), \tag{1}$$

where $v(t)$ is the angular speed of the motor. Hence, $\dot{v}(t) = dv(t)/dt$ represents the angular acceleration of the shaft of the motor. And $u(t)$ is the control input. The negative sign in Equation (1) is introduced to indicate the voltage polarity applied to the motor concerning the shaft rotation direction. For instance, if $u(t)$ is positive, then the motor rotates in the counterclockwise direction. And vice-versa. Therefore, by setting the control algorithm as:

$$u(t) = k_g sgn(v(t) + \beta), \tag{2}$$

where $sgn(\cdot)$ is the signum function, $k_g \in R_{>0}$ is the control gain, and $\beta \in R$ is a given constant value; the closed-loop system (1) and (2) yields:

$$\dot{v}(t) = -k_g sgn(v(t) + \beta). \tag{3}$$

To study the stability of the closed-loop system (3), let us use the next quadratic Lyapunov function:

$$V(t) = \frac{1}{2}(v(t) + \beta)^2. \tag{4}$$

Then, its time derivative along the system trajectories (3) produces:

$$\dot{V}(t) = -k_g|(v(t) + \beta)|. \tag{5}$$

The above expression relates to a negative definite function of the time derivative of the corresponding Lyapunov function. This implies that the closed-loop system (3) is asymptotically stable. The DC-Motor model in (1) is an ideal one, and mathematically speaking, asymptotic stability is assured. But, in the real control realization, and due to other realistic factors, such that friction, backlash, and inductive effects of the motor coil, then it is prognosticated that the closed-loop system is stable. That is, there exist positive constants $\beta_0$, and $t_0$ such that:

$$|v(t)| \leq \beta_0, \quad \forall t \geq t_0. \tag{6}$$

Therefore, and due to the chattering nature of the control law, the angular speed of the motor is commuting around the value of $-\beta$. This implies that the angular direction of the motor shaft in commuting too.

**Remark 1.** *The Lyapunov function (4) is a quadratic one. However, non-quadratic Lyapunov functions can be invoked to improve the control performance. This fact was evidenced in [23,24].*

To illustrate the controller performance from the simulation point of view, let us consider the system (1) and (2) with $k_g = 1$. Simulation results are shown in Figures 2 and 3. Here, we set $v(0) = 1$ m/s and $\beta = 0.1$. From these outcomes, it is observed that the performance of the controller is as expected. However, we have used a simple model of the DC-Motor. Hence, and to be more realistically, let now us the following DC-Motor with Coulomb friction:

$$\dot{v}(t) = -u(t) - f_c sgn(v(t)). \tag{7}$$

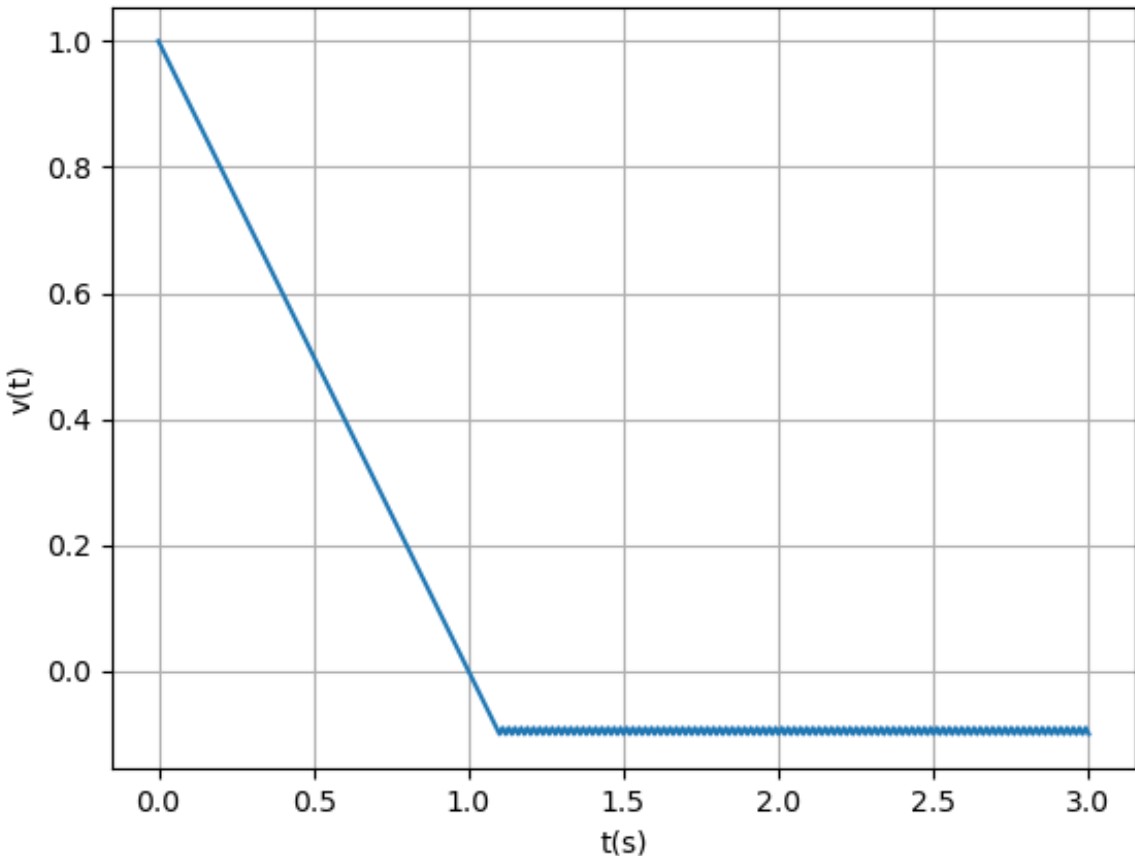

**Figure 2.** System output response.

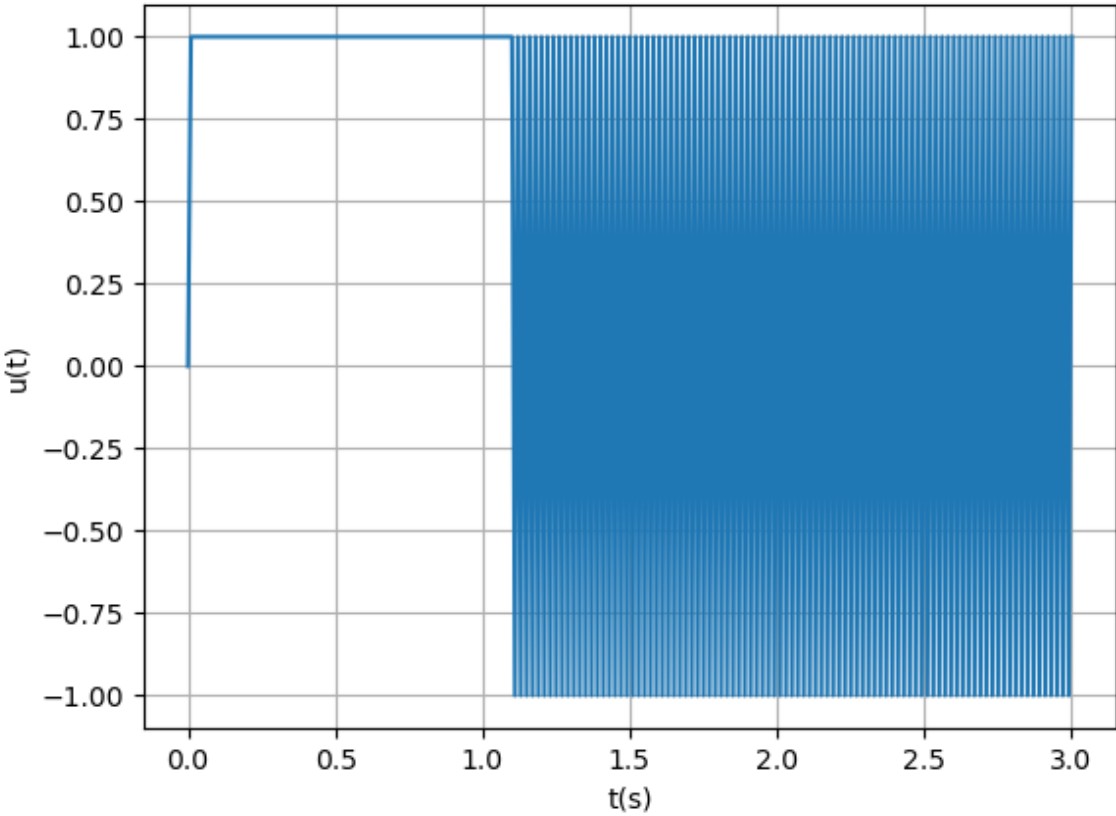

**Figure 3.** Resultant control command.

In the above expression, $f_c$ is the Coulomb friction parameter. Simulation results are now shown in Figures 4 and 5 by using the same initial condition and control gain than the previous example, but with $f_c = 0.2$. Finally, let us consider again the friction scenario but noisy in the feedback loop, that is:

$$\dot{v}(t) = -u(t) - f_c sgn(v(t)),$$

and

$$u(t) = k_g sgn(v(t) + \beta + N(t)),$$

with $f_c = 0.2$, $\beta = 0.1$, and $N(t)$. $N(t)$ is a normal (Gaussian) noise with standard deviation of 1.0 and mean equal to 0. Figures 6–8 show the corresponding numerical results. We can appreciate that the performance of the controller is as expected and stated in (6).

The previous numerical experiments were realized by using Python, and by invoking the first Euler approximation technique for the time-derivative. We set 0.01 second for the integration step in the numerical algorithm.

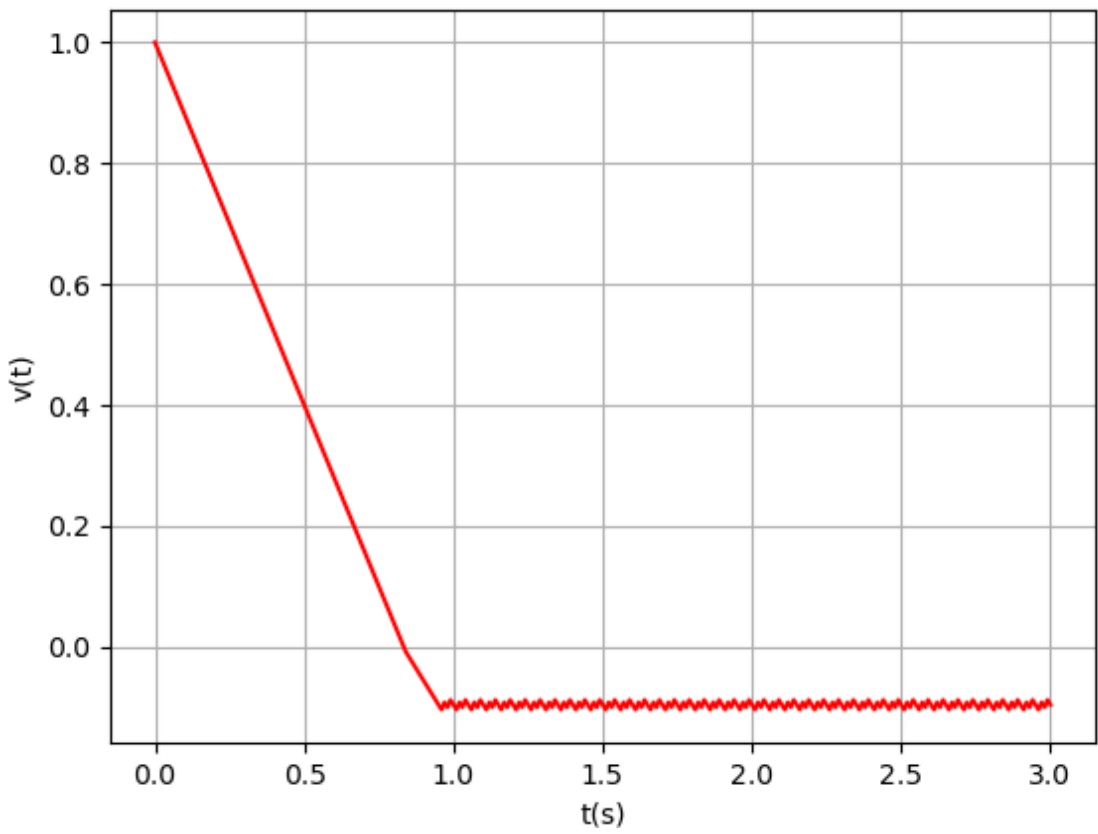

**Figure 4.** System output signal: perturbed case.

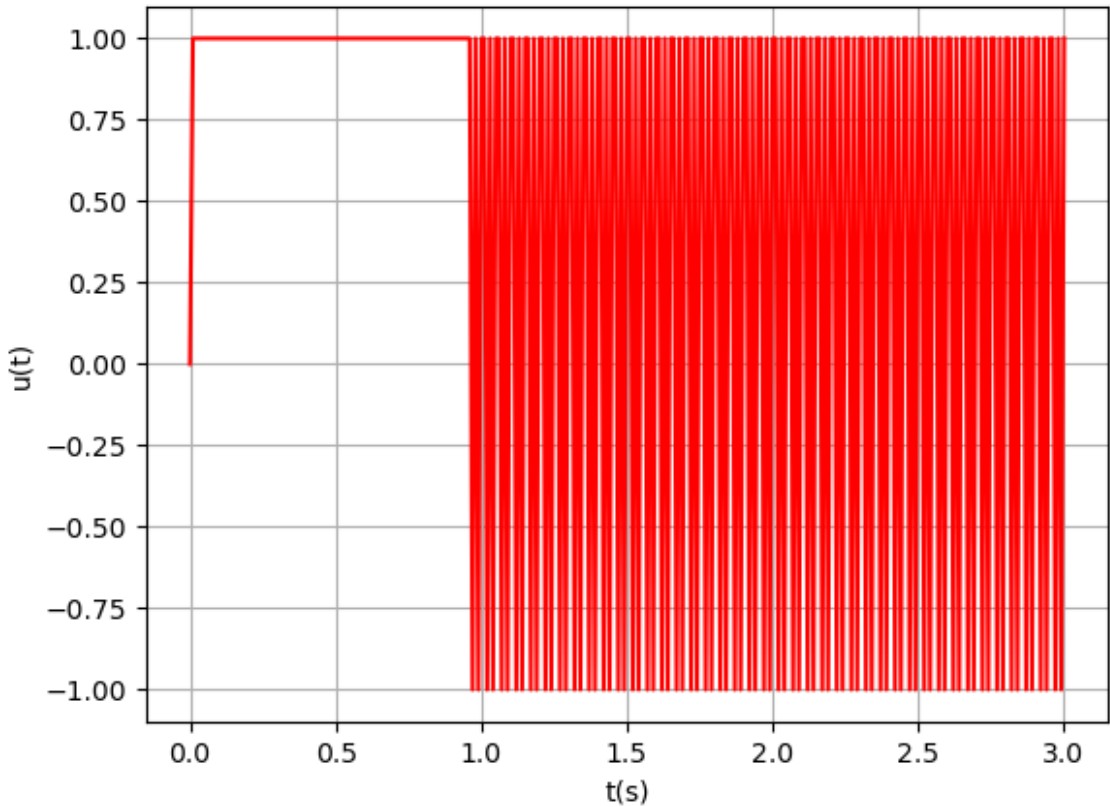

**Figure 5.** Resultant control signal: perturbed case.

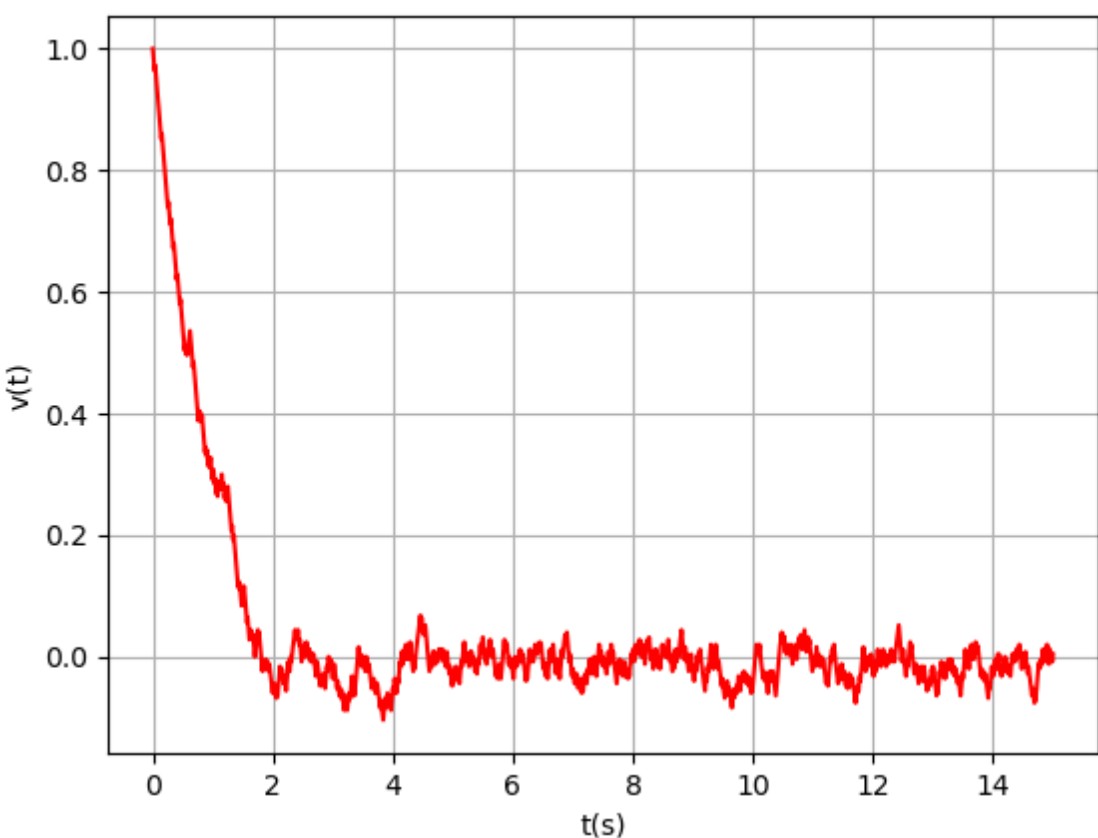

**Figure 6.** Resultant system output signal: sensor noisy case.

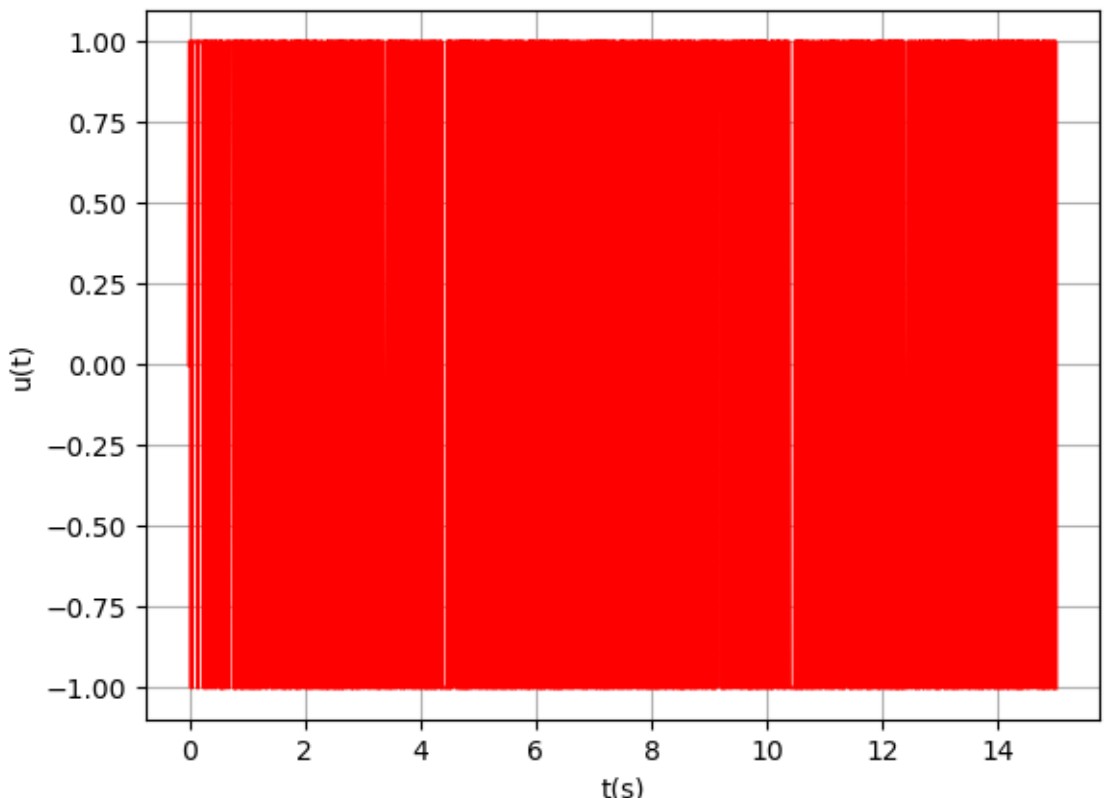

**Figure 7.** Resultant control signal: sensor noisy case.

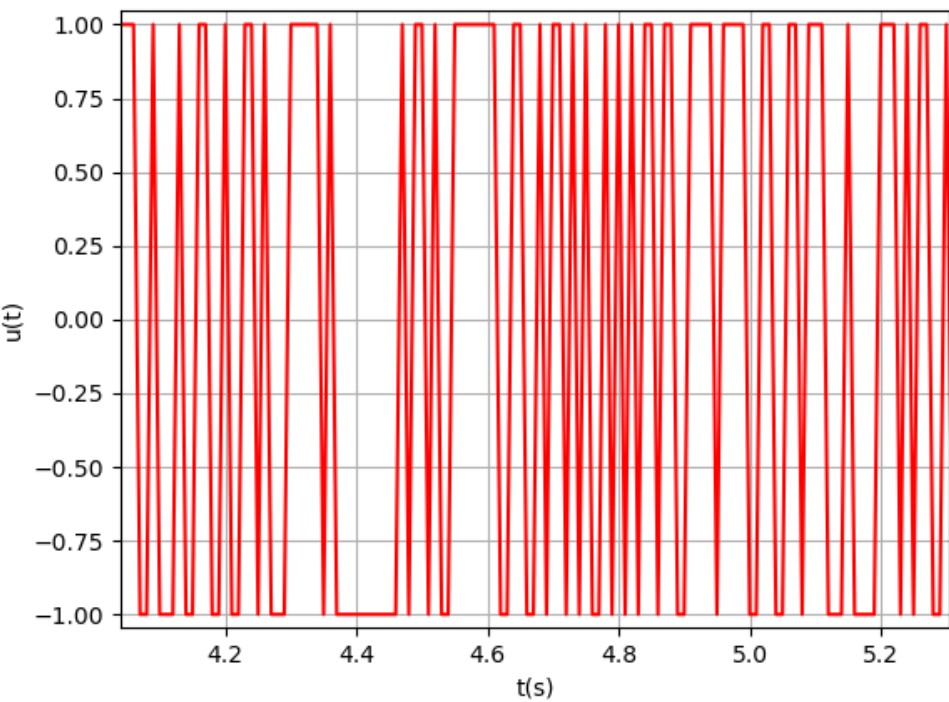

**Figure 8.** Resultant control signal: sensor noisy case, a zoom-in version of the previous picture.

## 3. Control Realization: Materials and Methods

This section presents the electronic realization of the control law stated in (2) and then connected to the DC-Motor. This is shown in Figure 9 by utilizing operational amplifiers (Opamp). Figure 10 displays a photo of the experimental platform of the cited closed-loop system.

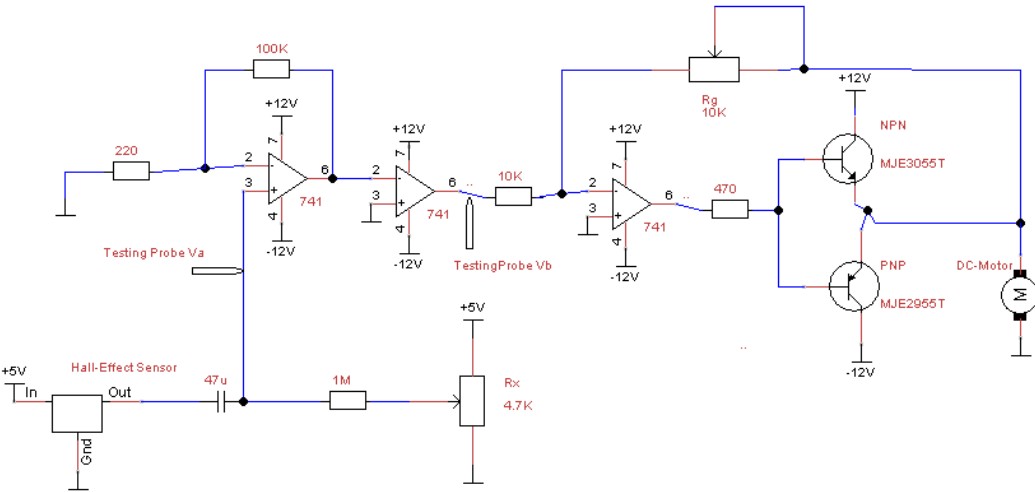

**Figure 9.** Electronic realization of the control scheme by employing opamps. Here, all electronic elements are in standard units. For instance, the resistor element 1 M represents a resistance of 1 MΩ, and the capacitor 47 μ a capacitance of 47μF, and so on.

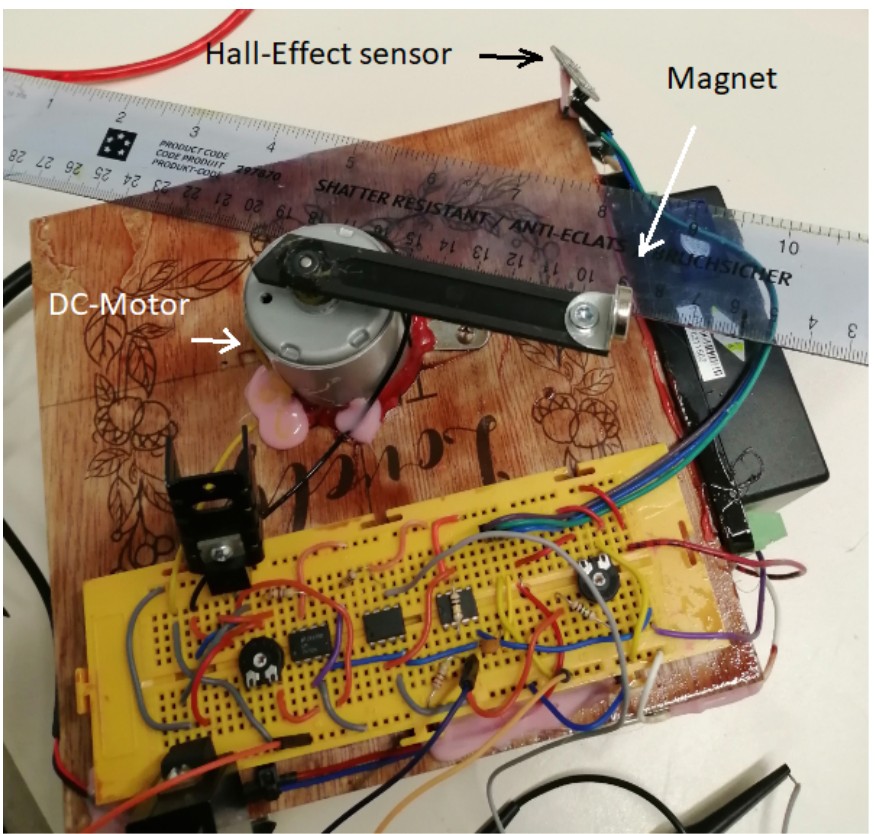

**Figure 10.** A low-cost experimental platform.

The electronic circuit statement can be obtained as follows. Due to our control algorithm (2) employs information of the rotational speed of the motor shaft, and under the chattering dynamic around a constant value, its measurement is realized by using time-derivation of the information supplied by the Hall-effect sensor. See the experimental video link https://www.youtube.com/watch?v=SMHZpda2W9w (accessed on 2 April 2021). Hence, the mission of the capacitor at the output of the magnetic sensor is precisely to carry on this differentiation task. And the potentiometer of $R_x = 4.7$ KΩ supplies the constant value $\beta$. Then, and by utilizing basic circuit analysis, we obtain that the voltage $v_a$, see the related testing probe in Figure 9, yields:

$$v_a(t) = (1 - \alpha)[5 + 0.221\alpha v(t)], \tag{8}$$

where $\alpha$ corresponds to the potentiometer manually adjusted which is moved from one extreme ($\alpha = 0$) to the other ($\alpha = 1$). Hence, $\alpha \in [0, 1]$. In consequence, the voltage at the testing probe $v_b$, see the related testing probe in Figure 9, results:

$$v_b(t) = -V_{sat}sgn[5 + 0.221\alpha v(t)], \tag{9}$$

being $V_{sat}$ the saturation voltage of the operational amplifiers. Finally, the resultant voltage at the terminals of the motor approximately is equal to:

$$u(t) = k_g sgn[5 + 0.221\alpha v(t)] = k_g sgn(v(t) + 22.6/\alpha). \tag{10}$$

The control gain $k_g \leq 1$ is adjusted by moving the potentiometer $R_g$. See Figure 9. Figures 11 and 12 show the obtained experimental results. In both Figures, we can observe a chattering behavior on the related signal from the control action. And for a perspective point of view, Figure 13 shows both pictures into one. Figure 14 is a zoom-in version of Figure 13.

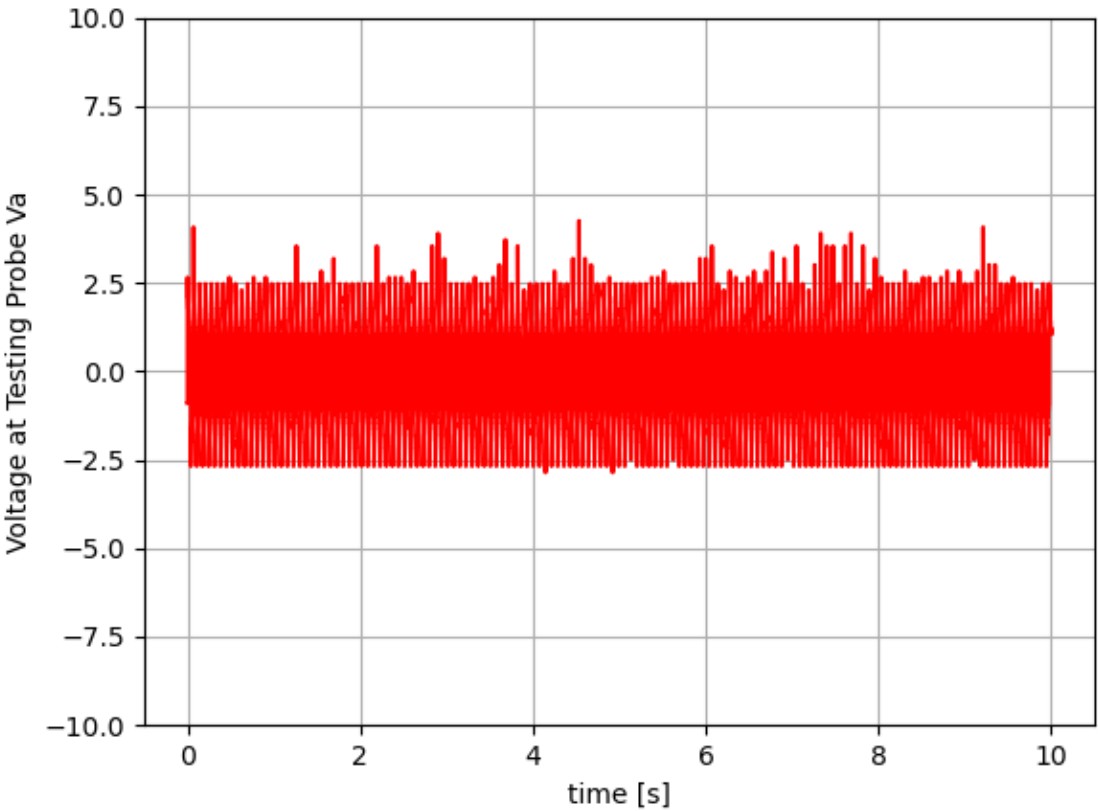

**Figure 11.** Experimental results.

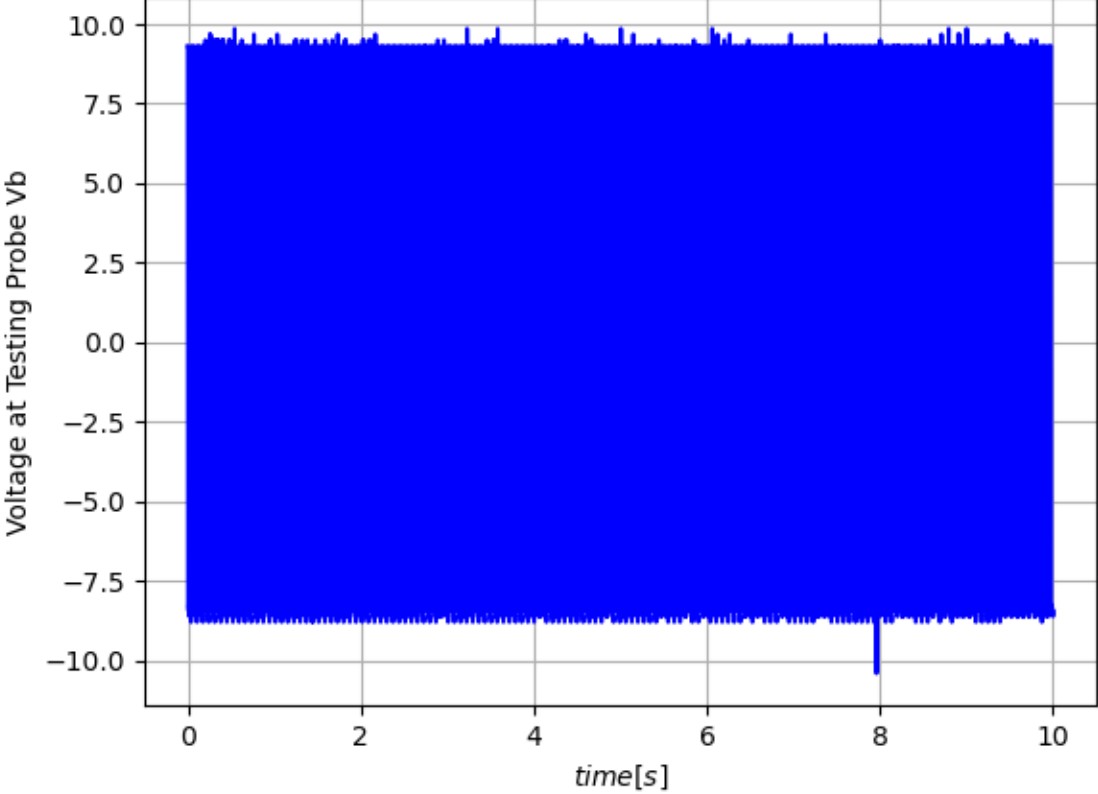

**Figure 12.** Experimental results.

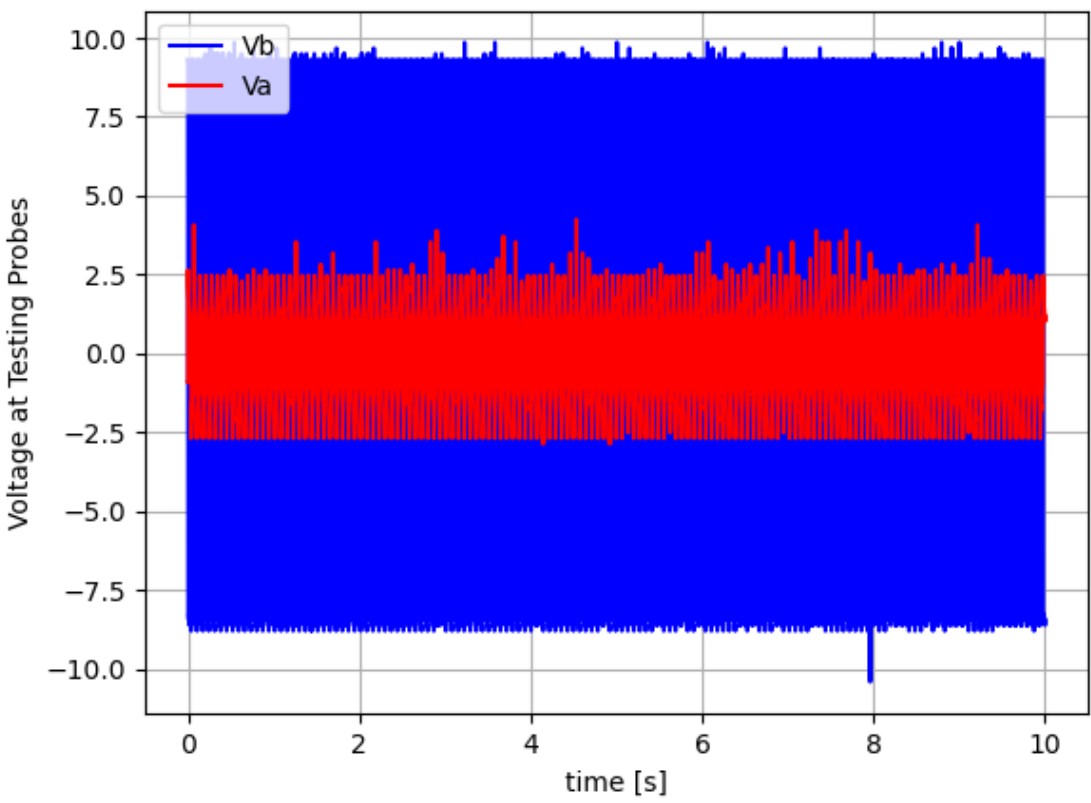

**Figure 13.** Experimental results for both signals, $v_a$ and $v_b$.

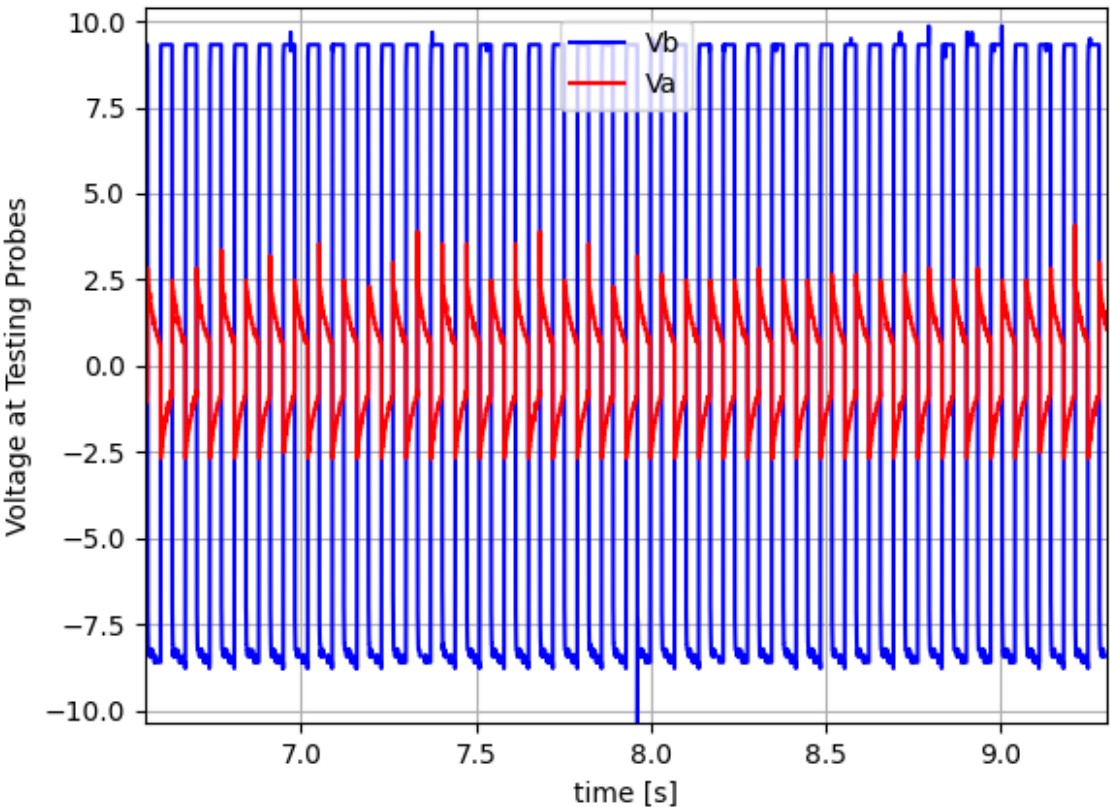

**Figure 14.** Experimental results for both signals, $v_a$ and $v_b$: a zoom-in version of the previous picture.

According to the video located at https://www.youtube.com/watch?v=SMHZpda2W9w (accessed on 2 April 2021), we can observe that the DC-Motor follows the moveable Hall-effect sensor. Even more, we can also note the vibrational reaction of the motor shaft whereas trying to follow the reference set-point marked by the cited sensor. Figure 15 shows a photo of the overall experimental platform on display.

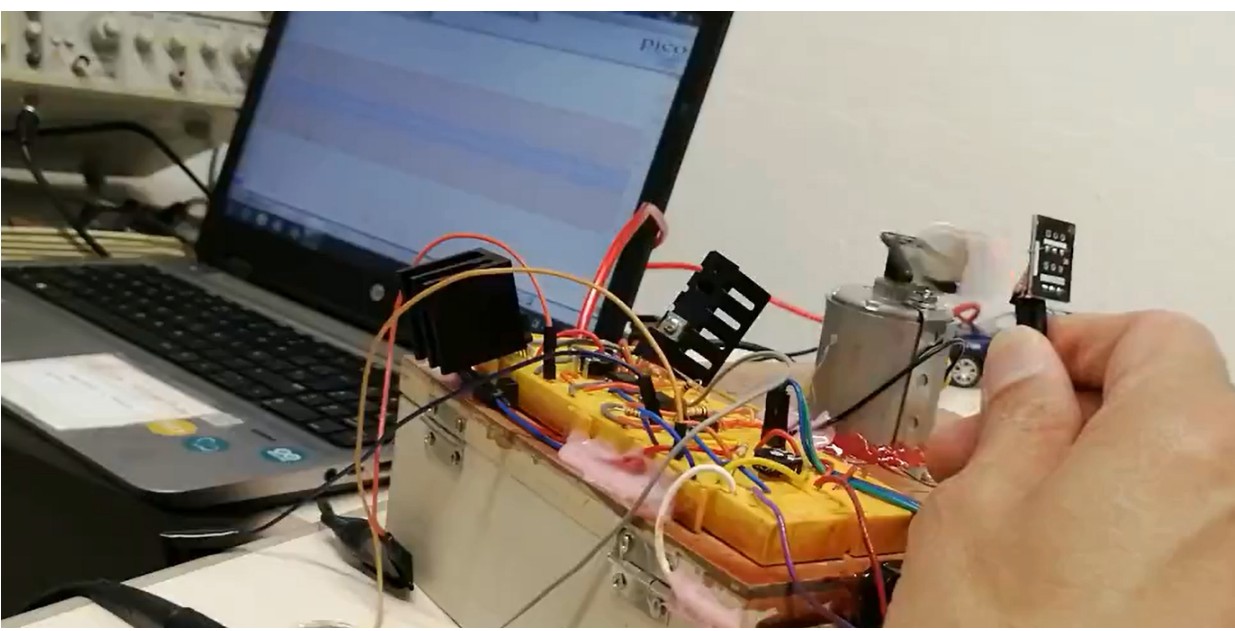

**Figure 15.** The experimental platform on the action.

## 4. Discussion

Our control approach is based on the vibrational control philosophy [25,26]. That is, vibrational control is applied by oscillating the control signal at a low amplitude and high-frequency [25,26]. Even more, this technique can stabilize unstable open-loop systems without feedback [25]. Vibrational controllers are in the range of the sliding-mode systems [18]. Therefore, our experimental platform may be considered as an experimental realization of a kind of sliding mode control method. Moreover, our vibrational control statement does not use position feedback but its derivative. This option was taken into account due to the desire to manipulate the DC-Motor externally. Besides, we consider that this experimental board can be used to test other control methods. Therefore, this platform can complete a set of well-known experimental devices, such as the inverted pendulum and the ball and beam experiments are [27–29].

To conclude finite-time convergence, let us analyze the closed-loop system stated in (3), and below replied but by using the signum function definition:

$$\dot{v}(t) = -k_g sgn(v(t) + \beta) = \begin{cases} -k_g, & \text{if } v(t) \geq -\beta \\ k_g, & \text{otherwise} \end{cases}. \tag{11}$$

Then, by integrating the previous mathematical equation, we produce:

$$v(t) = \begin{cases} -k_g t + v_0, & \text{if } v(t) \geq -\beta \\ k_g t + v_0, & \text{otherwise} \end{cases}, \tag{12}$$

where $v_0$ is the initial condition. Figure 16 shows a graphical representation of the last equation versus time. From this picture, it is clear that the closed-loop system trajectories converge, in finite-time, to $v(t) = -\beta$. A sliding-mode motion of the system trajectory takes place when it arrives at the sliding surface given by $v(t) = -\beta$. This motion can be

defined in the Filippov sense [18,30,31]. But, from Figure 16, if $k_g$ is strictly positive, then a solution exists at the sliding-surface, which, in real applications, it is translated as a zig-zag motion around the sliding-surface producing chattering [18]. See Figure 17. This precisely is the source of a vibrational control signal. Even more, and due to the simplicity of our system, and because we also know a solution to the closed-loop system dynamic trajectory stated in (12), we can estimate the reaching-time, $t_r$, to the sliding-surface as:

$$t_r = \frac{v_0 + \beta}{k_g}.$$

(13)

Therefore, the reaching-time will depend on the system initial condition, $k_g$, and $\beta$. Hence, if the control gain is increased, then the reaching-time is reduced. This is the same conclusion in some switching controllers [18,32]. So far, a question arises: why does the parameter $\beta$ is integrated into the control algorithm? Well, the key point is observed in the output configuration of the power stage in the electronic realization of the controller (see Figure 9). The push-pull power amplifier produces the so-called crossover-distortion [33], and it is nothing else than the dead-zone nonlinearity to the control signal. See Figures 18 and 19. And as an option to mitigate its effect on the plant, we introduced this parameter. Experimentally, if this parameter is set to zero, or no motion is induced on the motor, or the system starts violently losing stability. Moreover, and related to the mathematical model of the DC-Motor stated in (1). This model is conceived by just using the DC-Motor manufacturing point of view, and this is not enough accurate due to other factors involved in a real DC-Motor, such as friction, motor coil inductive effects, and backlash. And the main intention of vibrational control is to mitigate these non-linearities and to a possible exogenous perturbation too. Table 1 shows the estimated items cost and the total one for our experimental platform.

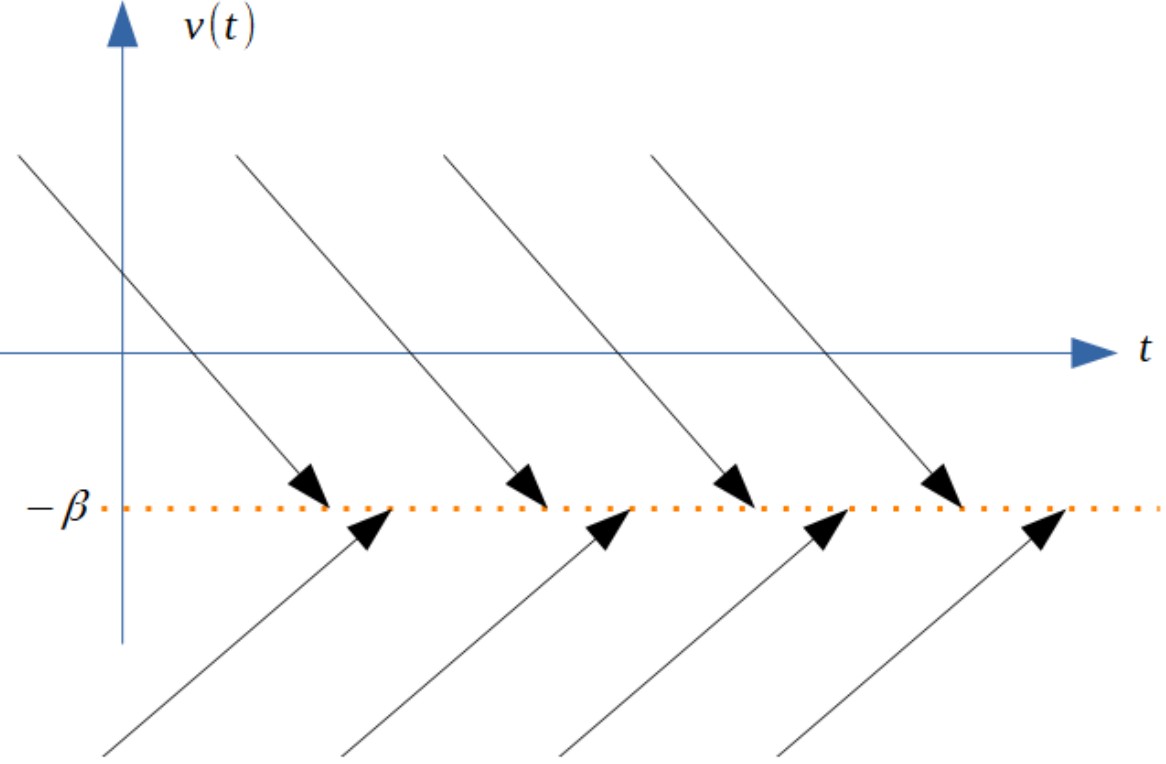

**Figure 16.** Phase trajectories of the closed-loop system (3).

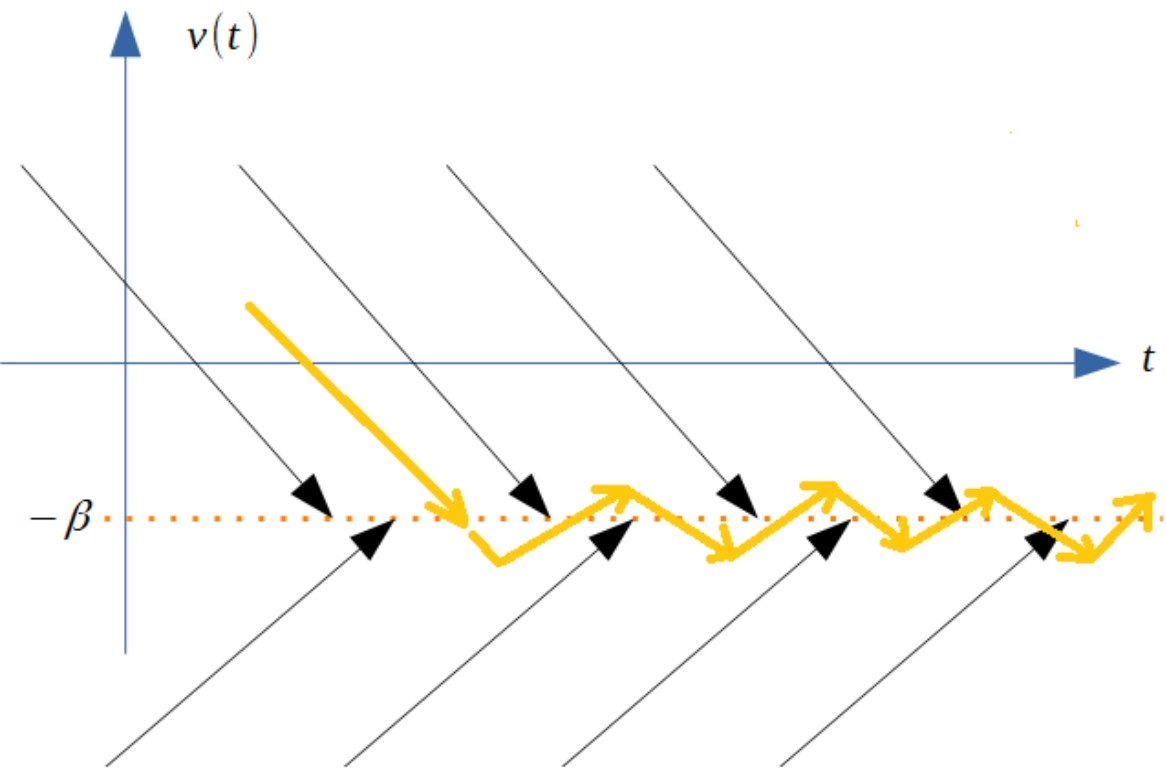

**Figure 17.** In a yellow line: A realistic trajectory of the closed-loop system (3) presenting chattering.

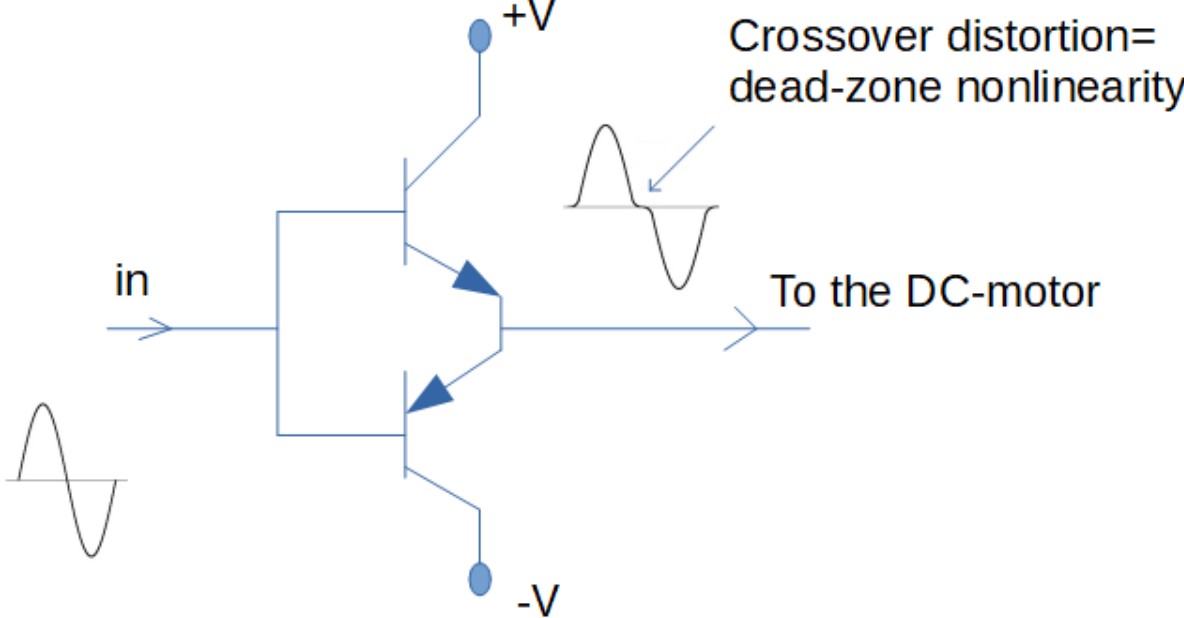

**Figure 18.** Illustration of the cross-over distortion which actually is the dead-zone nonlinearity.

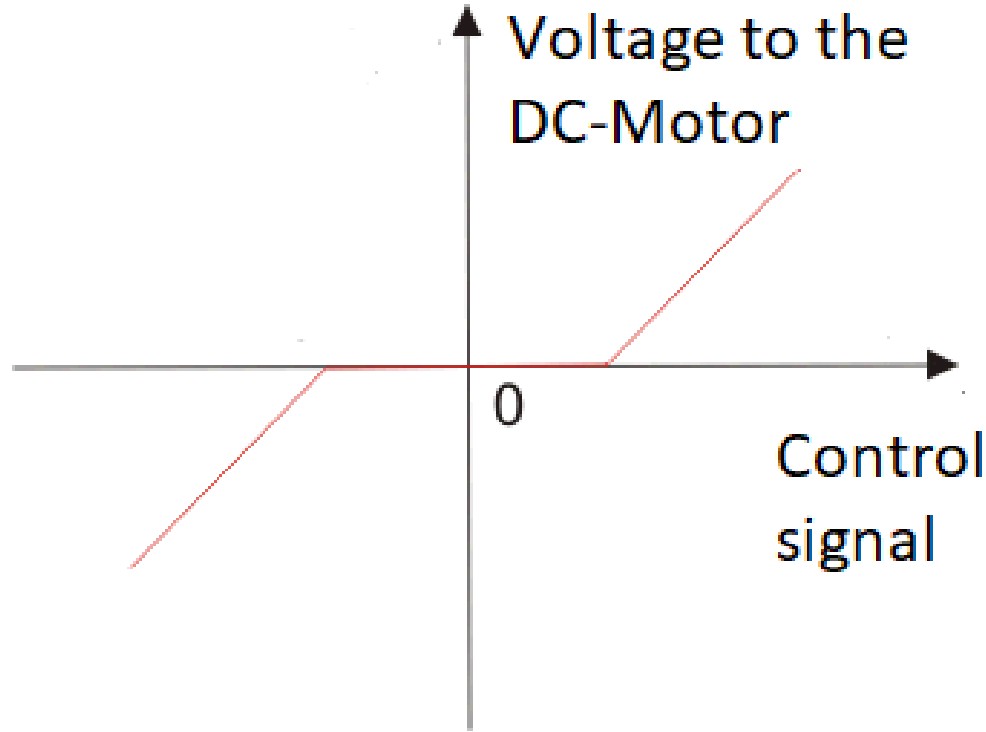

**Figure 19.** Actuator dead-zone nonlinearity.

**Table 1.** Electronic components list and estimation cost in Euros.

| Element | Specification | Price |
|---|---|---|
| 3 Operational amplifiers | LM741 | 3 |
| 2 Power transistors | MJE305T, MJE2955T | 4 |
| 2 Potentiometers | Mechanical | 1.5 |
| 1 Permanent magnet | | 0.50 |
| 1 DC-Motor | 12/24 Volts | 15 |
| 1 Bread board | | 4 |
| 1 Dual power supply | | 20 |
| 1 Hall-Effect sensor | | 0.50 |
| Others | | 0.50 |
| Total | | 50 |

Nowadays, the development of low-cost experimental platforms is now possible due to the accessibility and low-priced of the electronic components [34–36]. Finally, in the Appendix A, there is the reference Python code used in our numerical experiments.

### 5. Conclusions

As final comments, it is observed that the idea to externally manipulate a motor by employing vibrational control theory seems interesting to the academic field of control, instrumentation, and electronics. The granted video link on the experimental performance of our control approach evidences it. Furthermore, some important facts have been considered, such as the dead-zone nonlinearity. The main contributions of this paper can be summarized in the following items:

- A mathematical design of a vibrational control based on a simple model of a DC-Motor.

- A control scheme that strategically incorporates a constant parameter to mitigate the actuator dead-zone nonlinearity.
- A low-cost realization of the resultant controller.
- A contribution of a low-priced experimental platform to vibrational control.
- A control method that uses a magnetic Hall-effect sensor.

On the other hand, the main findings of this work can be summarized as follows: (1) external manipulation for position control of a DC-Motor can be realized by manipulating a moveable magnetic Hall-effect sensor; (2) vibrational control is acceptable for vibrational control of DC-Motors, and (3) an experimental control platform is realizable at low-cost.

**Funding:** This research received no external funding.

**Institutional Review Board Statement:** Not applicable.

**Informed Consent Statement:** Not applicable.

**Data Availability Statement:** Data sharing not applicable.

**Conflicts of Interest:** The author declares no conflict of interest.

**Abbreviations**

The following abbreviations are used in this manuscript:

DC　　　　Direct current
Opamp　　Operational amplifier

**Appendix A. Python Code**

This section presents the Python code used in numerical experiments.

```python
import numpy as np
import matplotlib.pyplot as plt
t=[0]
v=[1.0]
uc=[0]
h=0.01
k=0
kg=1
while(True):
    u=kg*np.sign(v[k]+0.1+np.random.normal(scale=1,loc=0.0))
    v.append(v[k]+h*(-u-0.2*np.sign(v[k])))
    t.append(h+k*h)
    uc.append(u)
    k=k+1
    if t[k]>15-h:
        break
plt.figure(1)
plt.plot(t,v,color='red')
plt.grid(True)
plt.xlabel('t(s)')
plt.ylabel('v(t)')
plt.figure(2)
plt.plot(t,uc,color='red')
plt.grid(True)
plt.xlabel('t(s)')
plt.ylabel('u(t)')
plt.show()
```

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
