# Peer review of "A Nonlinear Magnetic Stabilization Control Design for an Externally Manipulated DC Motor: An Academic Low-Cost Experimental Platform"

_machines, doi:10.3390/machines9050101_

Round 1

Reviewer 1 Report

  1. The English writing style and grammar should be improved.
  2. Even though the goal of the paper was to build a low-cost platform, it is not clear why the analogue electronic components are used.
  3. The main contribution of the paper seems to be the design of vibration controller for a positioning system containing DC motor. However, this control method is not discussed in the Introduction part in detail. The literature overview dealing this subject is insufficient.
  4. Figures 14-17 are excess, as no particular explanation in the text is referred to them.
  5. Controllers that are most often used for regulation of position and speed in electrical drives are linear PID controllers. These controllers are also simple and cheap solutions, so the advantages of proposed nonlinear controller are not clear.

Author Response

Dear reviewer, thank you very much for you help and patience on the written English. Below is a reply to each of your inputs comments:

1) “The English writing style and grammar should be improved.”

Response: Yes, I agree. Now, the written English quality of the paper has been improved. Thank you for your patience on the subject.

2) “Even though the goal of the paper was to build a low-cost platform, it is not clear why the analogue electronic components are used.”

Response: Now, this issue is mainly dealt in the first paragraph of the Introduction section. It follows from the fact that analog electronics do not require a programmer nor a computer as the low cost-micro_controller units do. Hence, analog electronics are cheaper and easy to employ.

3) “The main contribution of the paper seems to be the design of vibration controller for a positioning system containing DC motor. However, this control method is not discussed in the Introduction part in detail. The literature overview dealing this subject is insufficient.”

Response: In the introduction section we have motivated the paper contributions in contrast to other controllers on position control of DC-Motors, including the classical PID methods. And two other references were added too.

4) “Figures 14-17 are excess, as no particular explanation in the text is referred to them. Controllers that are most often used for regulation of position and speed in electrical drives are linear PID controllers. These controllers are also simple and cheap solutions, so the advantages of proposed nonlinear controller are not clear.”

Response: It depends on the visual appreciation to evidence by photograms of a controlled analog movement. The frame capture figures were liminated and a description of the video was added in in line 146-149. And related to PID controllers, we have added some remarks on it in the first paragraph of the Introduction section too.

Thank you for your inputs.

Reviewer 2 Report

Comments to the authors

Manuscript ID: machines-1183786

Title: A nonlinear magnetic stabilization control design to a DC motor and externally manipulated An academic low-cost experimental platform

1) The model shown in (1) is very simple. The author need to justify this model. Why is this model still accurate, despite its simplicity?

2) The Lyapunov function in (4) is quadratic. AS shown in 10.1016/j.sysconle.2014.12.001 and 10.1080/00207179608921661 a non-quadratic Lyapunov function usually yields better performance. Please use the above-mentioned references to discuss this very important point as a remark in the manuscript for interested readers.

3) The sign function in the control input causes the chattering in Figures 3 and 5. Explain why this chattering is acceptable in practice.

4) The conclusion section is too poor and should ne enhanced. It should properly summarize the main findings of the work.

Author Response

Dear reviewer, thank you very much for you help and patience on the written English. Below is a reply to each of your inputs comments:

“1) The model shown in (1) is very simple. The author need to justify this model. Why is this model still accurate, despite its simplicity?”

Response: This motivation is also described in the Discussion section. This model captures the basic DC-Motor behaviour. And due to its simplicity, and because the control design is based on the plant modelling,  the obtained controller becomes simple too. Other external factors, such as backlash and friction, are then mitigated by the chattering part of the control action.

“2) The Lyapunov function in (4) is quadratic. AS shown in 10.1016/j.sysconle.2014.12.001 and 10.1080/00207179608921661 a non-quadratic Lyapunov function usually yields better performance. Please use the above-mentioned references to discuss this very important point as a remark in the manuscript for interested readers.”

Response: Thank you. An important contribution. See Remark 1 on the new paper version.

“3) The sign function in the control input causes the chattering in Figures 3 and 5. Explain why this chattering is acceptable in practice.”

Response: In the Introduction section, a reference on the use of chattering controllers to DC-Motor in real applications is given.

“4) The conclusion section is too poor and should be enhanced. It should properly summarize the main findings of the work.”

Response: Thank you. It was improved. Please see line 215-219:

"On the other hand, the main findings of this work can be summarised as fallows. 1) external manipulation for position control of a DC-Motor can be realized by manipulating a movable magnetic Hall-effect sensor; 2) vibrational control is acceptable for vibrational control of DC-Motors, and 3) a new experimental platform has been conceived to test other controller approaches."

Thank you very much for your inputs.

Reviewer 3 Report

This paper presents a nonlinear control design applied to position control of a DC motor and externally manipulated. However, the motivation is not provided clearly, which makes the contribution of this paper confusing. Moreover, there are several flaws in the current version, and the issues that need to be addressed are as follows:

  1. The motivation of this paper should be further clarified. For example,
    • In the Introduction part, the author mentioned a recent view that the position control of a DC motor is manipulated by a movable Hall-effect magnetic sensor externally. Can the author add some explanations about the differences or advantages of this view compared to existing methods?
    • The Introduction part is too simple.
    • It is difficult for the reviewer to see the innovation/meaning of this paper from the title and Introduction.
  2. The function of parameter $\beta$ in Equation (12) can be briefly explained, which can correspond to the specific explanation in Section 4 and the contribution in the Conclusion part.
  3. Figures 14-17 may be deleted and replaced with sentences to remind readers to watch the video provided.
  4. Authors need to correct several grammar and reference formatting errors in the manuscript.

Author Response

Dear reviewer, thank you very much for you help and patience on the written English. Below is a reply to each of your inputs comments:

“This paper presents a nonlinear control design applied to position control of a DC motor and externally manipulated. However, the motivation is not provided clearly, which makes the contribution of this paper confusing. Moreover, there are several flaws in the current version, and the issues that need to be addressed are as follows:”

“1. The motivation of this paper should be further clarified. For example,

  • In the Introduction part, the author mentioned a recent view that the position control of a DC motor is manipulated by a movable Hall-effect magnetic sensor externally. Can the author add some explanations about the differences or advantages of this view compared to existing methods?
  • The Introduction part is too simple.
  • It is difficult for the reviewer to see the innovation/meaning of this paper from the title and Introduction.”

Response: The Introduction section was expanded to highlight the main paper contributions. On position control to DC-Motors, there are many contributions. However, two other references were added. But from the vibrational control point of view, and by using speed estimation from a movable Hall-effect sensor to realize a nonlinear position controller seems a recent approach, including its realization by analog electronics. 

“2. The function of parameter $\beta$ in Equation (12) can be briefly explained, which can correspond to the specific explanation in Section 4 and the contribution in the Conclusion part.”

Response: I have tried to reduce this explanation. But, I had the feeling that some key points would be missing to beginner readers on control and electronic lectures. Please, forgive me.

“3. Figures 14-17 may be deleted and replaced with sentences to remind readers to watch the video provided.”

Response: Figures 15-17 were deleted. Furthermore, a description of the video was added in line 146-149.

"According to the given video located at here, we can observer that the DC-Motor follows the movable Hall-effect sensor by hand. Even more, we can observe the vibrational reaction of the motor’s shaft on following the reference set-point marked by the cited sensor. Additionally, Figure 14 shows a photo of the overall experimental platform on action."

“4. Authors need to correct several grammar and reference formatting errors in the manuscript.”

Response: Thank you for your patience on the written English. It was improved, and the references revisited. 

Thank you.

Round 2

Reviewer 1 Report

1. The Introduction section is still inconsistent and does not provide a sufficient background for this paper. 
2. A programmer is not "a no low-cost electronic device".
3. Even though the author states that designing a digital controller requires programming knowledge, as well as reading a long manuals, I must notice that choosing proper analog electronics components also demands reading manuals, building electronic boards and soldering experience.
4. English writing style is still poor.

Author Response

Thank you for your inputs. The new paper version was prepared by following these suggestions. Written English was revised with the help of a supporter. On the other hand, the introduction of the paper was also expanded. We add five more references on low-cost experiments and teaching (see lines 62-71 of a new paragraph),  and modified the first and second paragraphs too. Other points on the induction were also adjusted and took into account your observation on the fact that the programmer is also a low-cost device.  

Thank you.

Reviewer 3 Report

I am satisfied with the revision and have no comments.

Author Response

Thank you. Written English of the paper was also improved with the help of a supporter. 

Best regards.